# Can Positive Mindsets Be Protective Against Stress and Isolation Experienced during the COVID-19 Pandemic? A Mixed Methods Approach to Understanding Emotional Health and Wellbeing Needs of Perinatal Women

**DOI:** 10.3390/ijerph18136958

**Published:** 2021-06-29

**Authors:** Jacqueline A. Davis, Lisa Y. Gibson, Natasha L. Bear, Amy L. Finlay-Jones, Jeneva L. Ohan, Desiree T. Silva, Susan L. Prescott

**Affiliations:** 1Telethon Kids Institute, 15 Hospital Avenue, Nedlands, WA 6009, Australia; Lisa.Gibson@telethonkids.org.au (L.Y.G.); amy.finlay-jones@telethonkids.org.au (A.L.F.-J.); jeneva.ohan@uwa.edu.au (J.L.O.); desiree.silva@telethonkids.org.au (D.T.S.); susan.prescott@telethonkids.org.au (S.L.P.); 2School of Medicine, The University of Western Australia, Perth, WA 6009, Australia; 3School of Public Health, Curtin University, Bentley, WA 6102, Australia; 4School of Medical and Health Sciences, Edith Cowan University, Perth, WA 6027, Australia; 5Institute for Health Research, Notre Dame University, Fremantle, WA 6160, Australia; natasha@bearstats.com.au; 6Joondalup Health Campus, Joondalup, WA 6027, Australia; 7inVIVO Planetary Health, Worldwide Universities Network (WUN), West New York, NJ 10704, USA

**Keywords:** COVID-19, perinatal, pregnancy, human isolation, loneliness, mental health, wellbeing, mindfulness, self-compassion, mixed methods, experiences and perceptions

## Abstract

The aim of this study was to explore the relationship between emotional health and wellbeing and support needs of perinatal women during the COVID-19 pandemic, and to understand their experiences and need for support. This is a potentially vulnerable group and a critical developmental phase for women and infants. A mixed methods design was used to collect quantitative and qualitative data that provided a robust insight into their unique needs. A total of 174 women who were either pregnant or post-birth participated. The main findings demonstrated that women in this cohort experienced varying levels of stress and isolation but also positive experiences. Exploring the relationship between mental health (perceived stress and wellbeing) and resilience (mindfulness and self-compassion) revealed an association between positive mental health and higher levels of mindfulness and self-compassion. Positive mindsets may be protective against psychological distress for the mother and her child, suggesting that meditation-based or similar training might help support expectant and post-birth mothers during times of crisis, such as a pandemic. This information could be used to make recommendations for future planning for practitioners and policymakers in preparing for prospective infection waves, pandemics, or natural disasters, and could be used to develop targeted tools, support, and care.

## 1. Introduction

The unprecedented impact of COVID-19 on mortality and morbidity and global upheaval has far exceeded that of any recent disease outbreak (World Health Organisation, 2021). The large-scale impacts of lockdown restrictions on social, emotional, and economic wellbeing are predicted to have unparalleled and extensive implications for mental health in broad populations, independent of biological effects of infection [1]. Unfortunately, containment efforts critical for halting the spread of the virus have increased social isolation, loneliness, relationship stress, and disconnection from communities [2]. To date, there is little understanding of the long-term impacts on mental health and wellbeing of a global pandemic of this scale, induced by lockdown restrictions, quarantine, physical distancing, loss of employment/income, and changes to lifestyle, particularly for vulnerable population groups. One such potentially ‘vulnerable group’ is women in the perinatal period, along with their developing child. Currently, little is known about the emotional health and wellbeing information and support needs of women expecting a baby during the COVID-19 pandemic and the barriers and enablers to that support.

Pregnancy, delivery, and the postnatal period can be a time of increased psychological distress [3]: up to 9% of women experience depression during pregnancy and up to 16% suffer from postnatal depression [4,5,6]. There is considerable evidence that psychological distress (stress, anxiety, and/or depression) during the perinatal period has detrimental effects on maternal health and can impact the long-term mother-infant relationship [7,8,9,10]. The known effects of stress during pregnancy on the growth and development of the foetus vary, but are among many adverse early life exposures that may have lifelong effects on the health and longevity of offspring, now well described in the Developmental Origins of Health and Disease (DOHaD) paradigm [11,12,13]. Accordingly, there is growing focus on optimising the early “exposome” (the totality of early exposures and experiences) as a critical opportunity to influence long-term resilience of individuals, families, and societies [14]. Environmental exposures, both detrimental and nourishing, manifest in personal and population health outcomes. Certain windows for health promotion interventions can be optimised through interactions between genes, the environment and time [14]. Protecting and nurturing maternal mental health is a critical public health issue, not only for mothers but for the next generation. In this context, it is important to understand and characterise the impact of the COVID-19 pandemic for this group in particular.

Numerous barriers already exist for women who attempt to access traditional perinatal wellbeing, psychological distress prevention, or treatment programs; in particular challenges in navigating psychosocial care systems [15]. Widespread restrictions imposed through the COVID-19 pandemic have generated additional barriers to accessing mental health and wellbeing information and services. Online interventions and support may be useful for women in the perinatal period during containment periods given the accessibility issues already faced by this population [16,17]. However, evidence-based online interventions are still not very accessible, widely disseminated, or well-integrated into perinatal health services.

The aim of this study was to explore the relationship between emotional health and wellbeing and support needs of perinatal women during the COVID-19 pandemic, and to understand their experiences and need for support. A secondary aim was to investigate if there were associations between individual resilience factors—namely, mindfulness and self-compassion—and mental health that could help perinatal women during a health crisis. We were also interested in any differences between pregnant women and those who had recently given birth. In this study we refer to ‘resilience’ as the ability to mentally or emotionally cope with a crisis or to return to pre-crisis status quickly. We view resilience traits as positive mindsets.

The ORIGINS Project (‘ORIGINS’) provided a valuable opportunity to address these questions. This is a decade-long collaborative initiative of 10,000 Western Australian (WA) families, enrolled during pregnancy and followed over the first five years of life. As a contemporary cohort study, ORIGINS is examining ways to optimise health potential of individuals and communities, going beyond disease prevention, to the conditions that facilitate flourishing from an early age—taking a broader approach to the protective and buffering factors that enhance resilience and reduce allostatic load [18]. In years to come this will also provide the capacity to assess the longitudinal impact of the COVID-19 pandemic on these young families, as well as strategies to mitigate this.

To address the initial impacts of the pandemic on this population, we used survey data collected in June and July 2020, during a lockdown period in Western Australia, together with follow-up interviews to deepen our understanding of the quantitative results. This was aimed at assessing the nature and magnitude of impact in ways that will help support this population in particular, and guide future healthcare planning and services.

## 2. Materials and Methods

For this study a mixed-methods sequential explanatory design was utilised with two distinct phases: analysis of our outcomes of interest in a sample of pregnant women drawn from the ORIGINS cohort, and qualitative exploration of women’s experiences and preferences in a stratified sample drawn from the cohort. A mixed methods approach was used, as in combination, quantitative, and qualitative methods complement each other and allow for a more robust analysis [19]. The qualitative data were collected through purposive sampling and analysed second in the sequence to elaborate on the quantitative results obtained in the first phase. The results of data from both phases were triangulated and interpreted together.

### 2.1. Participants

Eligible participants for the current study were perinatal women (i.e., pregnant women or women who had a baby) during 2020. Participants were drawn from a sample of women participating in an ORIGINS Project sub study (*n* = 461), the Community Wellbeing Project. All women were eligible to take part if they were expecting a baby in 2020 and had completed online questionnaires on their experience of living through the COVD-19 pandemic. A total of 174 (38%) women consented to take part in the current study and participated in the data collection analysed as the quantitative component. Follow up semi-structured interviews were conducted with a stratified sample of these 174 participants (*n* = 14), stratified by either high or low mindfulness and/or self-compassion traits as identified through self-report measures (top 20th percentile for mindfulness and/or self-compassion scores for “high” and bottom 20th percentile for mindfulness and/or self-compassion scores for “low”). These interviews were conducted to gather further data on the wellbeing information and support needs of ORIGINS pregnant women during the COVID-19 pandemic period.

### 2.2. Ethics

The project received ethical approval from the Ramsay Health Care WA I SA Human Research Ethics Committee (#2037). Prior to the commencement of the online questionnaire, information explaining the purpose and procedures of the study was provided, followed by a consent statement outlining the implications, risks and benefits of participation.

### 2.3. Procedure

As two sets of data were collected for this mixed methods study, we have separated and reported the quantitative data collection (Section 2.3.1) and qualitative data collection (Section 2.3.2) in their respective sequential orders.

#### 2.3.1. Quantitative Data Collection

During June and July 2020, participants in the ORIGINS Project were sent emails outlining this project with a link to the online questionnaires. Participants were given a two-week window to complete the questionnaire, with a text message reminder. Validated instruments with good psychometric properties were used to collect participant information, as well as a behavioural questionnaire on utilisation of services to support women’s emotional health needs (non-validated). The contextual behaviour questionnaire was developed in response to the immediate pandemic situation. As this was a unique circumstance, the questionnaire has not been tested for reliability. Participants were not stratified for the quantitative data collection.

##### Measurement Instruments

**Wellbeing****:** The Mental Health Continuum—Short Form (MHC-SF) [20] was used to measure mental wellbeing. The MHC-SF is a 14-item questionnaire that assesses three dimensions of positive mental health: emotional, psychological, and social wellbeing [21]. Each of the 14 items on the MHC-SF can be scored between 0 and 5; the total score on the scale can range from 0 to 70 points. Higher scores indicate a higher level of emotional wellbeing.

**Stress:** Perceived Stress Scale (PSS) (10 items) [22] is a brief scale and psychometrically robust measure to assess perceived stress of this cohort. The questions relate to feelings and thoughts in the last month. Norms for the PSS-10 in a US female population aged 18–29 years old were 14.2 (SD 6.2) and 13.0 (SD 6.2) for females aged 30–44 [23].

**Self-compassion:** Self-Compassion Short Form Scale (SCS) [24] is a 12-item on five-point Likert scale (0 = ‘Almost never’ to 5 = ‘Almost always’) to record how often one behaves kindly and caringly towards oneself in difficult life situations. Average scores for the Self-Compassion Scale are around 3.0, a score of 1–2.5 indicates low self-compassion, 2.5–3.5 indicates moderate, and 3.5–5.0 is an indication of high self-compassion [25].

**Mindfulness:** The Mindful Attention Awareness Scale (MAAS) assesses individual differences in the frequency of mindful states [26]. The scale is a 15-item (1–6 Likert scale) questionnaire to assess dispositional (or trait) mindfulness; a receptive state of mind in which attention, informed by a sensitive awareness of what is occurring in the present, simply observes what is taking place. The measurements from MAAS considers consciousness related to self-regulation and well-being. Based on a mean of all items, MAAS scores can range from 1 to 6.

**Contextual Behaviour:** A behavioural questionnaire was developed using an online platform (REDCap) to capture demographic information along with utilisation of services to support women’s emotional needs (non-validated instrument).

#### 2.3.2. Qualitative Data

Semi-structured interviews were conducted with a stratified subset sample of participants to gather further data on the wellbeing information and support needs of ORIGINS perinatal women during the COVID-19 pandemic period.

##### Stratification of Participants

Participants were stratified into the top and bottom 20th percentiles based on their scores on the MAAS and the SCS, as we were particularly interested in the women’s mindfulness and self-compassion attributes. The “high” group consisted of women who scored in the corresponding 20th percentiles as follows: high score in MAAS (top 20th percentile = high mindfulness) and high score in SCS (top 20th percentile = high self-compassion). The “low” group consisted of women at the opposite end of the scoring range, i.e., low score in MAAS and low score in SCS. Of these women in both the high and low groups, only those who indicated they were interested in a follow-up interview (*n* = 30) were contacted.

Participants were invited by text message to partake in a semi-structured interview by telephone or videoconference, depending on participants’ preferences. Participants were emailed brief information about the interview topic prior to the interview and informed they would be reimbursed $35 for their time. If women wished to participate, they provided online consent via an individual e-consent process. All interviews were recorded and transcribed with the consent of the participant.

##### Data Collection

A semi-structured interview topic guide was developed and piloted with two pregnant volunteers prior to data collection to refine the structure and flow of questions prior to undertaking the full interviews. Before commencing each interview, the interviewer clarified the aims of the study and provided a definition of emotional health and wellbeing. Interviewees were asked to reflect on their experiences during lockdown in particular (March to July 2020). There were four core interview questions, plus a list of sample probes for reference, centred around:How, and if, COVID-19 had impacted emotional health and wellbeing during pregnancy.The type of emotional health and wellbeing information and services participants accessed.Emotional health and wellbeing needs during the last few months.Their views on how the wellbeing needs of pregnant women best be supported during times of crisis (such as a pandemic).

All interviews were conducted by two interviewers (J.D. and L.G.; first and second authors) and transcribed verbatim. The interviewers met several times during the interviewing process to review core themes arising from the interviews. Interviews ceased in the absence of new themes, and once both authors agreed data had reached saturation point. None of the pilot data has been included in the final analysis.

### 2.4. Data Analysis

#### 2.4.1. Quantitative Data: Analytic Approach

Descriptive statistics were used to describe demographic information and utilisation of services to support women’s emotional needs (behavioural questionnaire) using means and standard deviations for continuous variables and frequency and percentage for categorical variables. Differences between pregnant and postnatal participants were assessed using independent t-tests for continuous data and Fisher’s exact test for categorical data. Scatter plots were produced to understand the relationship between variables, followed by a correlation matrix using Pearson’s correlations. Cronbach’s alpha (α) was calculated for the resilience and mental health scales to examine internal consistency. To examine the relationship between mental health (perceived stress and wellbeing) and resilience (mindfulness and self-compassion), Pearson’s correlations were calculated. Linear regression was used to further explore the relationships between mental health (perceived stress and wellbeing) and resilience (mindfulness and self-compassion). Univariable (unadjusted) models were produced followed by multivariable (adjusted) regression where the model was adjusted for self-compassion, mindfulness and pregnancy. R-squared terms are reported, representing the proportion of the variance explained by the predictors. All statistical analysis was performed using Stata v 16.1 (StataCorp, College Station, TX, USA).

#### 2.4.2. Qualitative Data

Qualitative data were analysed using qualitative content analysis assigning a code to each concept using NVivo. The Braun and Clarke approach to thematic analysis was undertaken and transcripts were iteratively coded, and codes collated into higher-level themes [27]. Data were analysed using a phenomenological approach (i.e., as a description of experiences as consciously experienced by participants) and narrative codes were deducted. Narrative codes were reviewed by both interviewers, with codes expanded or collapsed as required. Codes were collated into higher-level themes, using all data relevant to each theme. The themes were reviewed comprehensively for homogeneity by both authors before overarching themes were decided, with substantial supporting overlapping data. Differences in codes and themes between the “high” and “low” groups were recorded.

## 3. Results

### 3.1. Quantitative Statistical Analysis

#### 3.1.1. Sample

In total, 174 females completed questionnaires in June and July (Table A1, Appendix A); of those, 31 were currently pregnant and expecting a baby in 2020 and 143 were postpartum (i.e., up to one-year post-birth). The mean age was 33 years (SD 4.6) and the majority of the sample had at least tertiary education (i.e., 58% bachelor’s degree or above). Most of the sample had above average socio-economic status, with 64% in the least disadvantaged quintiles and 8% in the most disadvantaged quintile.

#### 3.1.2. Emotional Health and Wellbeing Scores

Descriptive statistics for the emotional wellbeing measures are presented in Table 1. We grouped measures into “resilience” (i.e., MAAS and SCS) and “mental health” (e.g., MHC-SF and PSS).

#### 3.1.3. Resilience (Mindfulness and Self-Compassion)

There was a significant difference in self-compassion between pregnant and postpartum women, with postpartum women scoring lower compared to pregnant women (mean (SD): postpartum 3.1 (0.8) vs. pregnant 3.6 (0.7), *p* = 0.003). There was no significant difference between pregnant and postpartum women for mindfulness (*p* = 0.169). Postpartum women had lower levels of self-compassion, compared to pregnant women (mean (SD): postpartum 3.1 (0.8) vs. pregnant 3.6 (0.7), *p* = 0.003). Resilience scales demonstrated good to excellent internal consistency (SCS, α = 0.89 and MAAS, α=0.93).

#### 3.1.4. Mental Health (Perceived Stress and Wellbeing)

There was a significant difference in perceived stress between pregnant and postpartum women, with postpartum women on average scoring a higher perceived stress score (mean (SD): postpartum 14.0 (6.6) vs. pregnant 10.5 (6.6), *p* = 0.009). There was no difference between pregnant and postpartum women for mental health (*p* = 0.086). Mental health scales demonstrated excellent internal consistency (MHCF α = 0.93 and PSS α=0.90).

#### 3.1.5. Associations between Variables

Unadjusted and adjusted models for mental health and perceived stress are shown in Table 2. The MHC was positively associated with SCS (*r* = 0.61) and MAAS (*r* = 0.59), such that higher levels of positive mental health were associated with higher levels of self-compassion and higher levels of mindfulness (Figure 1 and Figure 2). In the unadjusted MHC models, SCS and MAAS were associated (both *p* < 0.001), but pregnancy was not (*p* = 0.086). In the adjusted model both SCS and MAAS remained statistically significant where it was estimated that on average an increase in SCS of 1 resulted in an increase in mental health of 6.1 (95% CI: 3.9 to 8.3), and an increase in MAAS of 1 resulted in an increase in mental health of 4.7 (95% CI: 2.9 to 6.4). Forty-seven percent of the MHC variance was explained by the combination of SCS score and MAAS score (R^2^ = 0.47).

Perceived stress was negatively correlated with the SCS score (*r* = −0.63) and MAAS scores (*r* = −0.61), suggesting that higher levels of perceived stress were associated with lower levels of self-compassion and mindfulness (Figure 3 and Figure 4). Pregnancy was associated with the PSS in the unadjusted model, with pregnant women on average having lower levels of stress compared to postpartum women, however after adjusting for SCS and MAAS this relationship did not remain significant. In the unadjusted and adjusted PSS models, SCS and MAAS were statistically significant. In the adjusted model, an increase in SCS of 1 resulted in a decrease in PSS of 3.6 (95% CI: −4.8 to −2.3) and an increase in mindfulness of 1 resulted in a decrease in PSS of 2.6 (95% CI: −3.6 to −1.6). Forty-nine percent of the variance was explained by the combination of SCS score and MAAS score (R^2^ = 0.49).

#### 3.1.6. Information-Seeking and Utilisation of Services to Support Women’s Emotional Needs

Women reported seeking emotional wellbeing information from a variety of sources: 24% had received information from their workplace, receiving this weekly (40%) or every few weeks (45%). Furthermore, 46 participants (26%) reported seeking emotional wellbeing information from family and friends, predominantly every few weeks (39%). In total, 37 women (21%) had been in direct contact with a health professional during the previous few months, with 76% of these interactions in person rather than via telehealth (24%). Of those who used telehealth (n. 37), 16% found it effective. Only 11 women (6%) reported accessing online support for their emotional wellbeing, with four women using SmartApps and seven using websites. However, when asked the question, ‘*If you were to access online support, what type of support do you think you would be helpful?’*, 90 women (56%) indicated websites and 34 women (21%) indicated apps would be helpful. There were no significant differences in reference to the behavioural questions between pregnant and postpartum women (*p* > 0.05).

### 3.2. Qualitative

In total, 14 interviews were conducted with postnatal women between November 2020 and February 2021. Purposive sampling was used to achieve an equal distribution of participants from high and low resilience and mental health scores; there were 7 from the high group and 7 from the low group. All women were part of the ORIGINS Project and had given birth to a baby during 2020, with the majority postnatal at the time of interview. The mean age of the women was 33.8 (SD 3.5) years old.

Qualitative data was analysed using qualitative content analysis assigning a code to each concept. To ensure inter-rate reliability, narrative codes were reviewed by both interviewers, then collated into four overarching themes: 1. Impact of COVID-19 on psychological care; 2. Isolation from family and friends; 3. Information and support needs; 4. Positive outcomes (Figure 5). Differences within codes and themes between the “high” and “low” groups were recorded.

#### 3.2.1. Theme 1: Impact of COVID-19 on Psychological Care

(a)Psychological Distress:

Many women reported changes in their mental health and wellbeing since the start of the pandemic, in particular increased psychological distress (stress, anxiety, and/or depression). The effects were exacerbated for those with other young children to care for and those who experienced immediate restrictions in support services and access to family care.

“*I was really anxious. I did a calm birth course to try and help me just get some that sense of control back in a world where everything was so uncertain. So, I was just really stressed out … I just feel like pregnant women weren’t supported through the pandemic … You’re just thrown in the complete deep end because you don’t have regular support that you’d normally have*.”(CW6_low)

(b)Lack of Face-to-Face Service Access:

The greatest consternation from all women was the frustration with the lack of service access. However, this was more frequently reported by women in the ‘low’ group. Physical health services, such as ultrasound scans, took a long time or were replaced by telephone or telehealth consultations. Many of the women understood the rationale for this but reported that delays or cancellations heightened anxiety.

“*There was no face-to-face and just calling someone on the phone, it felt it was a bit impersonal. So, I relied heavily on my family to deal with my, you know, my anxiety and my outbursts and just being frightened for those few weeks*.”(CW1_high)

Some women recognised that they needed mental health support but were unsure how and where to access it.

“*It’s not until you’re in that deep, dark place that you need that help and someone can tell you, whereas if I’d known about it before, I might not have gotten to that point*.”(CW9_high)

There was also a sense among some of the women that by seeking care they were impacting on others.

“*… so I felt like I was taking away from other people by asking such a simple question but, really, it could have been a big deal for all I knew … I was not caring to myself, I was to my baby, I’m two people now and I needed that appointment just as much as anyone else*.”(CW9_high)

(c)Hospital Restrictions:

Women had varied experiences with the newly imposed hospital restrictions; generally, the high group were satisfied with the information and support from the hospital: “*I loved my experience there. I received heaps of support and it was really great*.” (CW9_high) Several women in the ‘low’ group struggled with the immediate impacts:

“*When they cancelled everything, it took weeks for them to do anything online and then it was pages of reading, there’s no videos, no nothing, and then you have all these questions, and you try to call to find out some answers and no one can give you any answers and there was no one to talk to*.”(CW4_low)

#### 3.2.2. Theme 2: Isolation from Family and Friends

(a)Social Isolation

Frequently women from both the high and the low groups mentioned their feelings of isolation, particularly from family but also from friends and peers.

“*It was quite sad that I couldn’t even share my pregnancy experience—as scary as it was, I couldn’t share that with anyone, and I feel like I missed out*.”(CW9_high)

“*There’s a whole group of mothers out there that are just there by themselves*.”(CW10_high)

(b)Lack of Birth Support Person

A major worry for women was potentially not being able to have a support person at medical appointments and at the birth. One participant in the ‘low’ group had to labour by herself and she strongly believed that this should not have been allowed. Even for women in the high group they emphasised the importance of a support person being there for the whole journey: “*To do it alone is terrifying*” (CW1_high).

#### 3.2.3. Theme 3: Need for Increased Information and Support Needs

(a)Increased Access to Child Health Nurses

The majority of women highlighted that they needed increased access to support, particularly face-to-face services. Most frequently, women discussed their needs to access Child Health Nurses (CHN) once they had their babies, as women believe they are important for mums’ emotional support as well as the child’s development. Some women expressed their frustration about delays in appointments, with one mum reporting delays of up to 2–3 weeks to see a CHN: “*That doesn’t help me when I’ve got something I’m worried about right now*.” (CW4_low).

“*For six months, there’s a whole group of mothers from late February to March/April that were just forgotten about and those mothers, some of them now at nine months make no connections with any other mothers because they were just left*.”(CW10_high)

Additionally, inability to physically check the baby exacerbated stress: “*… it’s a lot more stressful not knowing has the baby got enough weight or are they going okay or even just having that check-in*.” (CW11_low) “*I think when you’re pregnant, it’s very much that you want to show them physically what you’re concerned about*.” (CW14_low).

(b)Peer Support

Results suggested that peer support is fundamental to women at this time in their lives. Participants in our sample, in both the high and low groups, described that knowing that others are experiencing similar feelings, issues, and concerns provides a high level of comfort for women.

“*Just to be able to obviously talk to other people that were going through basically the same things just made it all a little bit easier*.”(CW4_low)

“*Having those people that we’re going through the exact same as what I was was really helpful*.”(CW14_low)

One of the participants (CW10_high) created a virtual mother’s group in response to the lack of peer connection. This has been very successful and there are over 200 members, some of whom now meet in person.

(c)Virtual Support

Women discussed a range of virtual supports they utilised during this period including telephone, videoconferencing (telehealth), social networks, online applications, text messaging, and videos. There were differing opinions on these support services; several women were quite positive about telehealth: “*Telehealth option is amazing.*” (CW6_low) However, frustrations were expressed about telephone appointments:

“*I didn’t have one successful phone call appointment*.”(CW14_low)

“*I*
*think that initial phone conversation when my anxiety levels are quite high, I just shut it down and then didn’t access services for a while*.”(CW11_low)

Online social groups were useful for some women, especially via Facebook, but others found the discussions overwhelming and had to shut them down: “*… limiting the access to social media as well, that would be one of the ongoing things to [my] wellbeing*.” (L_CW12). Few participants reported using online wellbeing applications, with the exception of one participant who used a mindfulness app almost every night, for “*helping wind down and getting to sleep*.” (CW9_high)

(d)Clear Information That Is Equitable and Easily Accessible

The lack of clear pregnancy communication was a source of considerable frustration for most of the women in both the high and low groups: “*I just feel like pregnant women weren’t supported through the pandemic.*” (CW6_low) However, there was recognition that blame could not be attributed to anyone in particular as it was an unprecedented situation. In responding to questions on how the wellbeing needs of pregnant women could best be supported during times of crisis, such as a pandemic, most of the women emphasised their reliance on family, friends, and other pregnant women:

“*… being able to speak to other women in the same situation and they have their coping mechanism and what they’re doing, that might help as well*.”(CW2_high)

One participant was especially aggravated by the lack of antenatal care information from the hospital so moved to a private hospital where she could access online information videos that met her care needs. However, she recognised the inequity in this situation:

“*You shouldn’t have to be going private to get this information because it’s really just standard information that anyone should be getting … you shouldn’t have to pay that amount of money to get what’s really a basic human right to understand and to get*.”(CW4_ow)

Information needs to be communicated that is easy to understand:


*[The information] … it’s just too complicated for a normal person to understand it. We’re not medics, we’re not in that field, so you’re just like, “What exactly are you trying to say?”*
(CW4_low)

Additionally, some information would have been preferable to no information:

“*It would have been useful to maybe have some generic information that went out to women in that situation, if you were pregnant, [for example], “It’s early days. We don’t know what the impact could be.” … Statements from a medical professional to put people’s minds at ease*.”(CW5_low)

#### 3.2.4. Theme 4: Positive Outcomes

(a)Time to Bond

Women in both the high and low groups expressed the benefits of less socialising and uninterrupted time to bond with their baby. Enforced social isolation positively contributed to several women’s general feelings of wellbeing.

“*I know a lot of us saw the benefits of a reduction in visitors … We were able to use coronavirus as an excuse just to stay in our little bubble*.”(CW1_high)

“*In fact, if anything I didn’t feel any pressure to go out and see people, I could just relax at home … I suppose the one good thing about not having everyone come over and visit was having that time to adjust to having a baby without lots of visitors and lots of expectations to host for people*.”(CW5_low)

“*It’s quite nice being less sociable … It’s quite nice not having all these different appointments booked in to see people. I think it just made everyone slow down a little bit*.”(CW11_low)

(b)Flexible Working Arrangements

Increased work flexibility was viewed as a positive aspect of the situation, both for women who had the opportunity to work from home during pregnancy and, especially, for their partners:

“*Working from home is quite a lot better and it definitely made my pregnancy a lot easier working from home*.”(CW11_low)

“*It’s been really great that my husband had been able to work from home a lot throughout the year … So, if that’s one thing that we could keep, would be him to continue to work from home*.”(CW3_high)

(c)Reaching a New ‘Normal’

Many participants drew positive aspects from their experience during the pandemic period, in both the high and low groups. Women adjusted their lives to the new situation, in particular their exercise routines:

“*I’ve made [exercise] more of a lifestyle that I just do every day rather than going to the gym or going to a class which I think the pandemic had made us do that kind of thing a bit more*.”(CW14_low)

For some women, particularly in the high group, the COVID pandemic did not appear to have impacted their experience of having a baby:

“*Everyone’s been wonderful. We’re just loving life at the moment. Yeah, absolutely. 2020 has been different, it’s had its moments, but overall, it’s been a great year for us*.”(CW3_high)

## 4. Discussion

The aim of this study was to explore the relationship between emotional health and wellbeing and support needs of perinatal women during a pandemic, and to understand their experiences and need for support. The main findings demonstrated that women in this cohort experienced varying levels of stress and isolation but also positive responses to the situation during and after the immediate impact of the COVID-19 restrictions in Western Australia. Overall, pregnant and postnatal women’s mental health (perceived stress and wellbeing) was similar to the general population in non-pandemic cohorts. Our qualitative findings were helpful in deepening and expanding on the quantitative outcomes. Interestingly, although women were stratified according to high and low mental wellbeing and positive mindset scores, very similar themes emerged between both groups.

Exploring the relationship with resilience mindsets (mindfulness and self-compassion) revealed an association between positive mental health and higher levels of mindfulness and self-compassion, suggesting there may be a modifiable pathway to improve resilience if these results are replicated in longitudinal and similar studies. Postpartum women had lower levels of self-compassion, compared to pregnant women, and there was a significant difference in perceived stress between pregnant and postpartum women, with postpartum women on average scoring a higher perceived stress score. This is understandable considering the immediate stressors associated with a new baby but does highlight the necessity to tailor information to these discrete groups. Generally, participants indicated they sought wellbeing information and support from a variety of sources but that they needed clearer, timely information that is more easily accessible to support their emotional needs.

Our research findings demonstrate that mental health is associated with self-compassion and mindfulness. Specifically, self-compassion and mindfulness predicted better mental health scores and lower perceived stress, consistent with other research in this field [28,29]. Although correlative, these findings indicate that self-compassion and mindfulness are potentially useful intervention targets for promoting mental health in perinatal women and supporting the development of compassionate traits in early child development [30]. A number of mindfulness programs have reported reduced rates of anxiety, depression, and stress during pregnancy [31]. Self-compassion has been associated with positive outcomes across a range of psychological functions, including lower depression severity, lower trait anxiety, and increased positive states [32]. The limited studies assessing Compassion Based Interventions (CBI) in the perinatal period have reported encouraging results [33,34]. The positive effects of meditation-based interventions, like mindfulness and self-compassion, may protect against anxiety in response to significant stress challenges, such as a pandemic, but more research is required.

Following a mixed-methods sequential explanatory design, our qualitative findings were helpful in deepening and expanding on the quantitative outcomes. Notably, although women were stratified according to high and low mental wellbeing and positive mindset scores, very similar themes emerged between both groups. Women in both groups reported experiencing stress and distress, particularly those in the low group, and both groups described support from family, friends, and peers as critically important at this time in their lives. Very few women used online supports, such as websites and apps, although women indicated they would find these mediums helpful. Inability to access services was described as frustrating, but women did manage to access some support in a virtual way. Interestingly, several participants, especially those in the high group, drew positives from their experiences such as more time to bond with the baby without pressure to socialise. These observations underscore the innate human capacity to adapt or adjust to new situations.

One of the key components of self-compassion is a sense of common humanity rather than feeling isolated during times of difficulty [35]. The importance of this was a theme echoed within our qualitative study: women took solace from knowing that others were in the same situation: “*Just to be able to obviously talk to other people that were going through basically the same things just made it all a little bit easier*” (CW4_low); “*Knowing that everyone was in a similar boat made it easier*” (CW14_low). Both mindfulness and compassion-based training interventions are receiving attention as they target negative symptoms and positive constructs; both can be taught effectively with minimal contact and online support [36,37]. Our findings from both the quantitative and qualitative data reinforce that positive mindsets could support perinatal women in times of crisis, along with the importance of timely, easy access to relevant support information that could be delivered in virtual platforms.

Due to the unique situation in Western Australia of low community transmission, our findings may not be generalisable to the rest of Australia or other countries. However, there are a number of conclusions that may be applicable to other settings. Firstly, our results showed an association between positive mindsets and better mental health, which suggests that these are protective in times of need, such as during a pandemic. Secondly, perinatal women need timely access to relevant, supportive information that can be provided in a virtual platform, but it should be appropriately targeted, and could be extended to family members, and other support networks. Thirdly, women are empowered through peer support, particularly when there is an emotional disconnect due to a lack of face-to-face interactions. Regardless of the setting, there is the potential that these simple implications could be adapted and applied by clinicians in a local context.

Although this study found associations between positive mindsets and better mental health during a crisis period, this was not appropriately designed to determine causality. Future controlled research studies are needed to determine if mindfulness and compassion-based training can prevent psychological distress during the perinatal period. In particular, web-based, minimal contact interventions could provide ready access to the resources needed during crisis periods. A current study is investigating the feasibility of a randomised controlled trial protocol comparing three meditation-based interventions delivered on the web during pregnancy [38]. If the study is shown to be feasible, results will be used to inform a future full-scale RCT in order to develop evidence-based, scalable, and low-cost interventions that promote wellbeing and reduce psychological distress among pregnant women.

A further limitation of this study was the homogeneous characteristics of participants: education levels were high, there was limited ethnic diversity, and above average socio-economic status. Therefore, our findings for this study are limited in their generalisability to marginalised groups. Future research should include a broader, more diverse, population cohort.

This study was strengthened by the use of a mixed method design. In conducting quantitative data collection in the first phase, subsequently we were able to stratify participants between “high” and “low” mental wellbeing groups, thus enabling qualitative data collection of comparative experiences between women who had higher positive mindsets and those who had lower positive mindsets. Our interpretation of both sets of data was more robust than exploring either set in isolation, in particular findings such as use of virtual platforms for wellbeing (websites, mobile apps etc). While women were receptive to using them (79%), in reality they had used them minimally (6%). On discussion, they experienced frustration with the complexity of information, conflicting information (social networks), or uncertainty of where to find credible information. This provides enormous potential to develop and target appropriate, accessible, low-cost evidence-based communications to pregnant women.

A key strength of this study was that it was nested within the infrastructure of a longitudinal birth cohort study, the ORIGINS Project [39]. Globally, ORIGINS is unique in its design [18], with synergies and benefits of multiple interventions and outcome data within the cohort. Importantly, the structure of ORIGINS enables agility and flexibility to rapidly embed research studies, responsive to real-world situations. As such, this study was nested within ORIGINS almost in ‘real-time’ with data collection occurring in the month after The World Health Organisation (WHO) declared COVID-19 a pandemic (11th March 2020). Understanding how psychological distress may, or may not, change across an unprecedented global pandemic is important for informing current prevention and intervention efforts, as well as planning for longer-term supports [2]. As the situation continues to evolve, the research team can continue to collect data on ORIGINS participants to determine conditions that are protective and facilitate flourishing from an early age.

## 5. Conclusions

Overall, the findings highlight the unique emotional wellbeing needs of women in pregnancy and post-birth during a crisis situation, such as the COVID-19 pandemic. This is in a contextual setting that, comparative to the rest of the world, has been relatively unimpacted by the pandemic, whereas the mental health impacts of perinatal women in most other parts of the world will be amplified by this crisis. Perinatal women are a unique and potentially vulnerable group and need targeted information and support. We conclude that resilience traits and positive mindsets may be protective against psychological distress for the mother and her child, suggesting that meditation-based or similar training for expectant women might help support them during times of crisis, such as a pandemic, but further research is needed. In sum, this information could be used to make recommendations for future planning for practitioners and policymakers in preparing for prospective infection waves, pandemics, or natural disasters, and is useful to develop future tools, support, and care. Our findings contribute to the wider literature on our understanding of the mental health impacts associated with a pandemic, specifically during and immediately after pregnancy.

## Figures and Tables

**Figure 1 ijerph-18-06958-f001:**
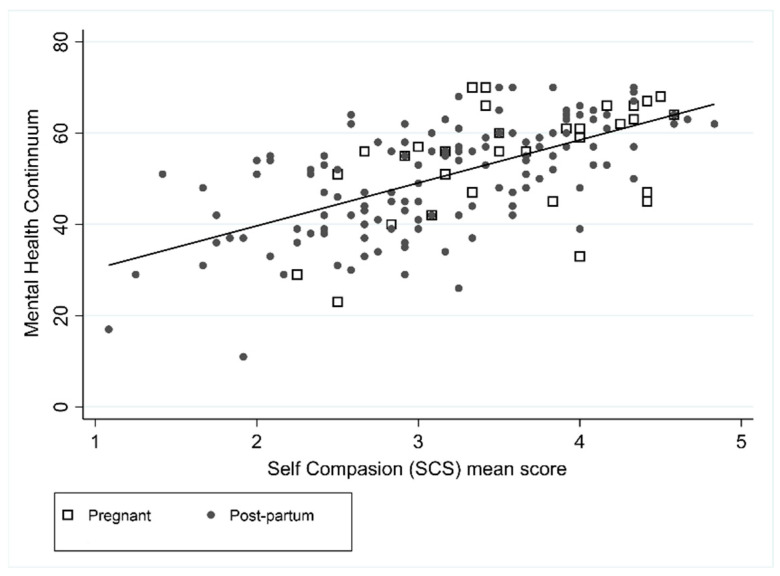
Association between MHC and SCS.

**Figure 2 ijerph-18-06958-f002:**
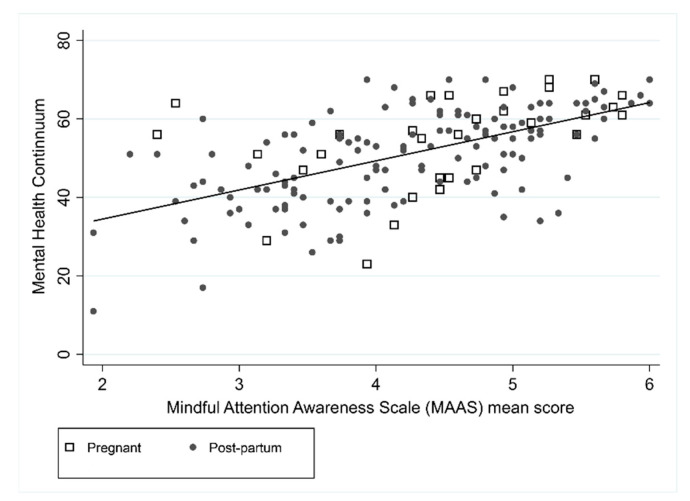
Association between MHC and MAAS.

**Figure 3 ijerph-18-06958-f003:**
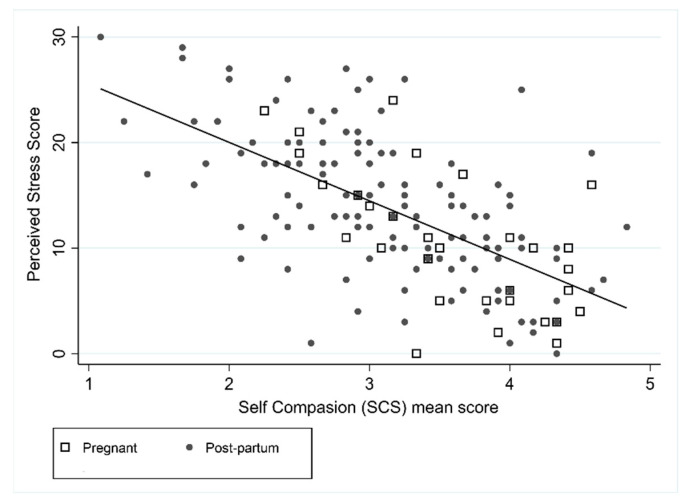
Association between PSS and SCS.

**Figure 4 ijerph-18-06958-f004:**
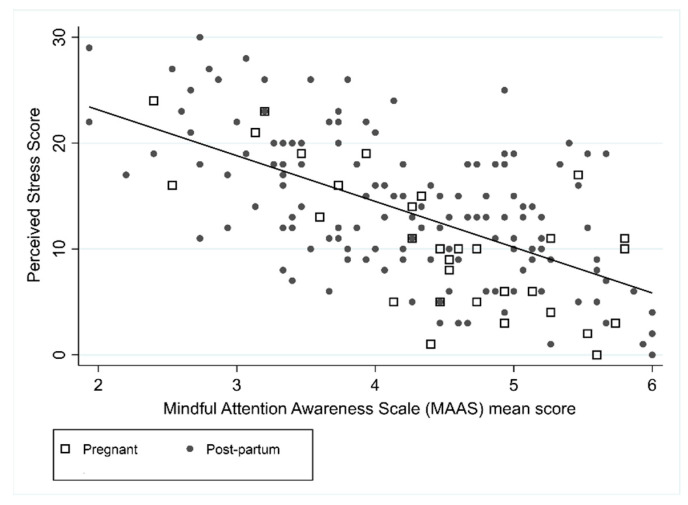
Association between PSS and MAAS.

**Figure 5 ijerph-18-06958-f005:**
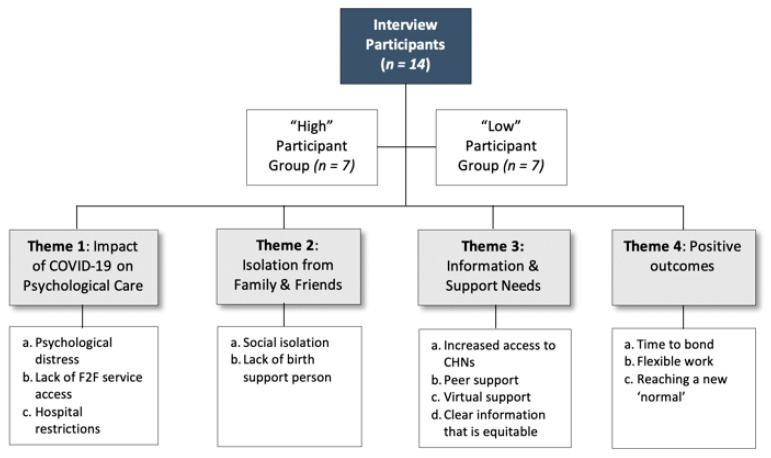
Key Qualitative Themes from both the “High” and “Low” Groups.

**Table 1 ijerph-18-06958-t001:** Summary statistics for all outcome measures.

Measurement Instruments	Postnatal	Pregnant	Total	*p*-Value
**Mindful Attention Awareness Scale (MAAS) Mean Score**	*n* = 139	*n* = 31	*n* = 170	0.169
Mean (SD)	4.2 (0.9)	4.5 (0.9)	4.3 (0.9)	
Range	1.9, 6.0	2.4, 5.8	1.9, 6.0	
**Self-Compassion Scale (SCS) Total Mean Score**	*n* = 136	*n* = 31	*n* = 167	0.003
Mean (SD)	3.1 (0.8)	3.6 (0.7)	3.2 (0.8)	
Range	1.1, 4.8	2.3, 4.6	1.1, 4.8	
**Perceived Stress Scale (PSS) Total Score**	*n* = 141	*n* = 31	*n* = 172	0.009
Mean (SD)	14.0 (6.6)	10.5 (6.6)	13.4 (6.7)	
Range	0.0, 30.0	0.0, 24.0	0.0, 30.0	
**Mental Health Continuum (MHC-SF) Total Score**	*n* = 140	*n* = 31	*n* = 171	0.086
Mean (SD)	50.5 (11.8)	54.6 (12.1)	51.3 (11.9)	
Range	11.0, 70.0	23.0, 70.0	11.0, 70.0	

**Table 2 ijerph-18-06958-t002:** Mental health (PSS and MHC) and its relationship with resilience (SCS and MAAS) and pregnancy using univariable and multivariable regression.

	Univariable (Unadjusted) Models			
	Mental Health Continuum (MHC)		Perceived Stress Scale (PSS)	
	Coefficient (95% CI)	*p* Value	Coefficient (95% CI)	*p* Value
**Self-compassion (SCS)**	9.4 (7.5, 11.3)	*p* < 0.001	−5.5 (−6.6, −4.5)	*p* < 0.001
**Intercept**	20.8 (14.6, 27.0)	*p* < 0.001	31.1 (27.6, 34.6)	*p* < 0.001
**Mindfulness (MAAS)**	7.4 (5.9, 9.0)	*p* < 0.001	−4.3 (−5.2, −3.5)	*p* < 0.001
**Intercept**	19.6 (12.8, 26.4)	*p* < 0.001	31.8 (30.0, 35.6)	*p* < 0.001
**Pregnant (Yes/No)**	4.1 (−0.6, 8.7)	*p* = 0.086	−3.5 (−6.0, −0.9)	*p* = 0.009
**Intercept**	50.5 (48.6, 52.5)	*p* < 0.001	14.0 (12.9, 15.1)	*p* < 0.001
	**Multivariable (Adjusted) Model**			
	**Mental Health Continuum (MHC)**		**Perceived Stress Scale (PSS)**	
	**Coefficient** **(95% CI)**	***p*** **Value**	**Coefficient** **(95% CI)**	***p*** **Value**
**Self-compassion (SCS)**	6.1 (3.9, 8.3)	*p* < 0.001	−3.6 (−4.8, −2.3)	*p* < 0.001
**Mindfulness (MAAS)**	4.7 (2.9, 6.4)	*p* < 0.001	−2.6 (−3.6, −1.6)	*p* < 0.001
**Pregnant (Yes)**	0.3 (−3.3, 3.8)	*p* = 0.885	−1.0 (−3.0, 0.9)	*p* = 0.302
**Intercept**	11.4 (4.7, 18.2)	*p* = 0.001	36.2 (32.4, 39.9)	*p* < 0.001

## Data Availability

The non-identifiable data presented in this study are available on reasonable request from the corresponding authors.

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
