# Peer review of "Can Positive Mindsets Be Protective Against Stress and Isolation Experienced during the COVID-19 Pandemic? A Mixed Methods Approach to Understanding Emotional Health and Wellbeing Needs of Perinatal Women"

_ijerph, 2021, doi:10.3390/ijerph18136958_

Round 1

Reviewer 1 Report

A few minor suggested revisions are indicated below.

  • At line 62, the authors refer to the exposome as an opportunity to influence "long-term resilience." It might be helpful if the authors would provide context on how/if resilience was defined/used in the design of this study.
  • The authors appear to use the terms resilience factors and protective factors (or buffering factors) interchangeably. The authors might consider offering a distinction for the use of each of these terms.
  • In lines 251 and 254, the authors describe mindfulness and self compassion as resilience outcomes/traits, but in the conclusion (line 612) describe these traits as protective factors.  The authors might consider correcting lines 251 and 254 to better reflect statements made in the conclusion.

Reviewer 2 Report

(1)The authors can add a separate paragraph to illustrate the limitations of the study.For example:the representativeness of this sample,the Ethnicity and educational level,whether the data were hierarchical in nature.Meanwhile, the authors should point out the future research directions.

(2)In the present study, the Cronbach’s alpha of the scale should be added.

(3)

Reviewer 3 Report

The introduction and discussion sections are sufficient and generally appropriate, but the methods and results need work.

The division of quantitative and qualitative methods induces confusion. For example, the authors state that the sample was stratified several times, starting on line 114. At this point, the reader is interested in how you stratified the sample. Rather than stating that a sample of participants had follow up interviews based on “high or low mindfulness and/or self compassion traits”, the authors should list the specific criteria (Was this determined by self-report scales and if so, what was the cut off points used for “high” vs. “low”?). However, this discussion does not occur until the “qualitative data” subsection. It would be helpful to follow typical JARS standards for reporting methods: participants (demographics, sample size, inclusion/exclusion criteria, ethics approval), measures, then procedure. The procedure section should tell the story of the study from the participants’ perspective.

Please report more about the contextual behavior questionnaire, including interitem reliability.

Please move Min/Max out of table 2 or create two subcolumns (range and mean/SD)

The use of the statistical methods appears to be inconsistent with the research question. The current t-tests in Table 2 ask the question, does trait mindfulness and self-compassion differ between women who have given birth and those who are currently pregnant. Is there a reason for this analysis? If so, please describe.

In Table 2, please remove “mean score (/6)” This information is redundant to what you have already described in the measures section.

Why are you categorizing the self-compassion scale? Categorizing a continuous variable reduces power in an already low sample size. Furthermore, this information is redundant to the preceding t-test.

Please describe the difference between the adjusted and unadjusted models and the justification for these analyses.

Table 3 appears to be more than just correlations, please revise the title as appropriate.

Is Table 3 supposed to be a regression table? If so, following standards for the journal will help readers better understand this table.

The qualitative data are very interesting. How were themes determined? What was the inter-rater reliability in coding themes?
